# A Review of Urban Microclimate Research Based on CiteSpace and VOSviewer Analysis

**DOI:** 10.3390/ijerph19084741

**Published:** 2022-04-14

**Authors:** Jiajing Li, Yang Mao, Jingyi Ouyang, Shuanning Zheng

**Affiliations:** 1Key Laboratory of Urban Environment and Health, Institute of Urban Environment, Chinese Academy of Sciences, Xiamen 361021, China; jiajingli@iue.ac.cn (J.L.); ymao@iue.ac.cn (Y.M.); jyouyang@iue.ac.cn (J.O.); 2College of Life Science, Fujian Agriculture and Forestry University, Fuzhou 350002, China; 3University of Chinese Academy of Sciences, Beijing 100049, China

**Keywords:** urban microclimate, research hotspots, research frontier, bibliometrics, scientometric

## Abstract

Urban microclimate has a direct impact on the quality of life of urban residents. Therefore, research on urban microclimates has received greater attention from contemporary scholars. At present, there is a lack of quantitative summary and review of the research in the field of urban microclimate, and it is urgent to sort out its research context and evolution. The Web of Science was used as the data source, and CiteSpace and VOSviewer software were used to analyze the urban microclimate research from 1980 to 2020. We discussed the annual trends, research countries, research institutions, key authors, highly cited publications, hot issues, and research fronts. The study found that: (1) the number of published articles on urban microclimate has experienced three stages: initial stage—slow growth period—rapid growth period; (2) European and American countries were the first to focus on urban microclimate research, while China started late but developed rapidly; (3) the research topics of urban microclimate are thermal comfort, improvement strategies, urban street canyons, and urban heat island effect; (4) the frontiers of urban microclimate include research on urban microclimate and building energy, ecosystem services, and urban parks.

## 1. Introduction

Urbanization is accelerating, which has caused large areas of natural surfaces to be occupied and replaced by artificial environments built by humans. Under such processes, urban industrialization becomes more intensified, and urban populations rapidly increase. Industrial production and population surges produce a large amount of heat emissions, and the dense building complexes exacerbate poor ventilation within the city. The heat island effect that results will lead to urban climate anomalies that increase energy and water consumption and affect the near-stratigraphic atmospheric circulation, causing more severe air pollution. This effect has different degrees of negative impact on urban biophenology and ecological processes. Among all the impacts that the heat island effect can have on cities, the urban climate is closely related to residents’ health risks and quality of life. Residents in high-temperature areas are more likely to suffer from physical discomfort, insomnia, and heat cramps than those in non-high-temperature areas [1]. However, China’s urban design and planning implementation strategies for alleviating urban diseases and improving microclimate discomfort are non-existent, lack scientific support, or are without decision-making significance. Therefore, research in the field of urban climate in China needs to be strengthened [2]. At present, people cannot control the climate on a large scale (such as global climate or regional climate based on local climate related to natural conditions), so when designing and building urban planning, it is necessary to consider how to better adapt to the above macro-climatic conditions. Compared with the large-scale climate, the microclimate in small-scale spaces (such as the block scale) can be influenced to a large extent by urban planners through various artificial means [3]. The microclimate is a small-scale climatic environment in the near-surface atmosphere and surface soil caused by different features of the underlying surface [1]. According to the classification of urban climate scales by Ooka and Shi et al., urban microclimate is a climatic indicator that describes a city within 1 km in the horizontal dimension and 120 m in the vertical dimension [4,5]. The urban microclimate is very important to people’s daily life. The significance of studying the urban microclimate lies in its ability to guide urban designers to create a more conducive public activity space by arranging site landscape elements and activity spaces [6]. The deterioration of urban microclimate will lead to increased urban energy consumption, increased pollution, and denser urban infrastructure, which will seriously affect the quality of life of urban residents [7,8,9].

At present, the research on urban microclimates mainly focuses on numerical simulation, the impact of building energy consumption, and the improvement measures of the microclimate. There is a lack of comprehensive and systematic quantitative summary and review. It is urgent to sort out the context and evolution of microclimate research to help relevant researchers have a more in-depth understanding of the current status and future trends of urban microclimate research. In this study, we explore research hotspots and future research development trends and provide a reference for future theoretical and practical research on urban microclimate.

## 2. Data and Methods

In this paper, the core database of Web of Science was selected as the data source. The search conditions were the subjects “microclimate”, “urban”, “urban micro-climate”, or “urban microclimate”. The document type was set as “Article” and “Review”, and the time range was set from “1990 to 2020”. In total, 1893 pieces of literature were retrieved.

This paper uses data mining, scientific measurement, information analysis, and other means for visual analysis to create a scientific knowledge map in the field of urban microclimates. Scientometrics is a modern theory that visualizes the development process of complex scientific knowledge, the overall knowledge framework, and its structural relationships by combining common methods of scientometrics with visualization techniques [10]. It is an emerging interdisciplinary scientific field. Taking a knowledge domain as the object, it displays the potential relationship network and development context of literature knowledge in related fields through literature text data mining, analysis, classification, mapping, and sorting. In this way it expresses the development process and structural relationship of scientific knowledge [11]. VOSviewer is a bibliometric analysis software jointly developed by Leiden University scholars Nees Jan van Eck and Ludo Waltman for drawing knowledge maps. It can be used for co-word analysis, co-citation analysis, and literature coupling analysis. It can display research results visually and has unique advantages in clustering technology, map displays, etc. [12]. CiteSpace is an information visualization software developed by Professor Chen Chaomei of Computer and Information Science from Drexel University in the United States [13]. The software directly, accurately, and quickly calculates the scientific field, analyzes the research trend in a certain field, and displays the research structure of the field in the form of a multi-integrated visual knowledge map by many researchers. Compared with traditional bibliometric methods, the visual analysis of scientific knowledge graphs is more intuitive and readable. This paper uses VOSviewer and CiteSpace software to visualize and analyze urban microclimate-related literature on the Web of Science from 1990 to 2020.

## 3. Results

### 3.1. Temporal Distribution of Publications

The temporal changes in the number of publications of a research field can reflect the pace of development in that field. Through CiteSpace software checking and screening, 1455 pieces of literature related to urban microclimate were obtained. The remaining 38 articles were duplicates and were therefore excluded.

Judging from the time distribution of the number of published papers, the research on urban microclimates at home and abroad can be roughly divided into three stages. (1) From 1990 to 2005, as the initial stage, there were very few studies on urban microclimate, mainly due to relatively weak cognition of the urban microclimate; (2) The period from 2006 to 2013 was a period of slow development, and the number of relevant documents increased year by year. Traditional urban climatology discusses the overall climate change and impact on the city [14], while architectural climatology is concerned with the creation and improvement of the indoor environment [15]. However, the urban microclimate research between the two developed slowly. It was not until the 20th century that the urban microclimate was paid more attention by experts and scholars (Figure 1). (3) From 2014 to 2020, we saw a period of rapid growth, and more than 1000 papers were published during this six-year period. Under the background of increasing global climate change, urban microclimate research was of great significance to the comfort of human settlements. In addition, with the United Nations climate change Paris Agreement reached in 2015, urban microclimate research has received greater attention. Therefore, the related research literature on urban microclimates increased rapidly.

### 3.2. Main Research Country or Region

To analyze the national cooperation network in the field of urban microclimate, the software CiteSpace was adopted. Country cooperation networks can reveal the distribution of research forces in the field of urban microclimate, and countries with frequent activities have strong research capabilities. As seen in Table 1, the research on urban microclimate began in western countries, but it has attracted global attention. Urban microclimate research is widely distributed around the world, and important countries engaged in urban microclimate research include Italy, Germany, the United Kingdom, Australia, Greece, France, and Canada, among others. The United States and China contributed the most, with around 300 publications each. Betweenness centrality refers to the measure of the extent to which a point is “in the middle” of other “pairs of points” in a graph. Notably, it can be seen from Figure 2 that the nodes in the United States, Germany, the United Kingdom, France, China, Italy, Malaysia, and some other countries that the betweenness centrality is above 0.1. This shows that these countries are important exchange hubs in the cooperation network. For example, scholars in the United States and Germany have conducted collaborative research on the topic of how urban morphology and urban design affect urban microclimates; authors from Italy and the United States have conducted collaborative research on the positive effects of urban vegetation on an urban and a building scale [16,17]. As urban microclimate research has become a global research hotspot, regionalization and globalization will become important trends in urban microclimate research. However, the participation of developing countries in the field of urban microclimate is still low, except in China.

### 3.3. Major Research Institutions

The contributions of each research institution are assessed by the research institutions affiliated with the authors. The cooperative network of research institutions can reveal the distribution of research forces in the field of urban microclimate, and the research institutions with frequent activities have strong research capabilities. Data in Table 2 and Figure 3 indicate that Chinese institutions rank sixth, showing that China pays attention to the field of urban microclimate. The report of the 19th National Congress of the Communist Party of China pointed out that China has guided international cooperation on climate change and become an important participant, contributor, and leader in the construction of global ecological civilization [18]. There are many institutions in China that study urban microclimates, but only the University of Hong Kong has reached 0.1 in the centrality of Institutional Intermediaries in China. China’s influence in the field of urban microclimates is relatively weak. Regarding China’s microclimate research, there remains a need to strengthen cooperation between different institutions and deepen the research from multiple perspectives. Arizona State University in the United States ranks first in terms of volume and intermediary centrality and shows that Arizona State University has the most obvious purple outer circle, as shown in Figure 3. Monash University has only published 13 papers but has a high centrality, showing that its published academic papers have important academic value. For example, scholars at Arizona State University have collaborated with scholars from the University of Colorado Denver, the University of Colorado Boulder, and the Technical University of Berlin on the relationship between urban morphology and urban microclimate [19,20].

### 3.4. Key Authors

A visual analysis of the authors included in the literature involved a total of 4684 authors out of 1455 articles, of which the most published author was Mattheos Santamouris of the University of New South Wales, who published a total of 18 articles. The key authors in the field of urban microclimate research are counted according to Price’s law [21]. The formula is as follows:(1)$$m_{p}\;=0.749\;\times \;\sqrt{n_{pmax}}$$
where m_p_ is the minimum number of papers published by core authors in the statistical period; *n_pmax_* is the number of papers published by the authors with the largest number of articles in the statistical period. Through calculation, m_p_ = 3.18, and 65 core authors with publications ≥ 4 were counted. Table 3 lists authors with publications greater than 9. With the help of VOSviewer to generate a commonly cited knowledge graph of authors, as shown in Figure 4, 1 node in the figure represents one author. The font size of the node depends on its weight; the larger the node, the more frequently the article is cited; the connection between the nodes represents the cooperative relationship between the authors. Through the knowledge graph of authors in a certain field, it is possible to understand the influence of authors in the field and analyze the research developments. As shown in Figure 4, the current research authors, and authors in the field of urban microclimate have less contact with other authors, forming multiple clusters of authors with a single lead author. Cluster 1 contains models and tools for climatology and biometrics represented by Andreas Matzarakis, such as RayMan, SkyHelios, CTIS, etc. Cluster 2, represented by Anna Laura Pisello, contains studies based on wearable sensing technology. Cluster 3, represented by Ariane Middel, contains studies on climate-sensitive urban forms, design, landscape, and infrastructure. Cluster 4, represented by Marjorie Musy, contains studies on urban microclimate and environmental performance modeling to design more sustainable urban forms, Cluster 5, represented by Robert D. Brown and C. Y. Jim, contains studies on urban microclimate design to make places more comfortable. Among them, Professor Andreas Matzarakis’ team (cluster 1) has a great influence, absorbing many core authors, and effectively contacting several research teams such as Anna Laura Pisello and Mohammad Taleghani to collaborate with each other. However, the cooperation between the various research teams is relatively scattered, and the connection is not yet close.

### 3.5. Analysis of Highly Cited Publications

The most cited publications in the field of urban microclimate and related information from 1990 to 2020 can be seen in Table 4. The most cited document is “Urban greening to cool towns and cities: A systematic review of the empirical evidence”, published in the journal LANDSCAPE AND URBAN PLANNING in 2010 and cited 1038 times. This paper systematically assesses the available evidence that greening measures (e.g., planting trees, building parks, or green roofs) affect the temperature in urban areas and points out that in order to effectively guide the planning and design of urban green spaces, the impact of the abundance, distribution, and type of greenery on temperature needs to be further studied [22]. In the top 20 highly cited papers, Eleftheria Alexandria [23], Edward Ng [7], Argiro Dimoudi [24], Toby N Carlson [25], and others have also begun to study the effects of green vegetation on urban microclimates.

“Applications of a universal thermal index: physiological equivalent temperature” and “Simulating surface-plant-air interactions inside urban environments with a three-dimensional numerical model”, were cited 535 and 487 times, respectively, ranking second and third most cited. The former describes physiological equivalent temperature (PET) suitable for assessing the heat composition of different climates [26]. Physiological equivalent temperature was proposed by German scholars in 1999 based on the thermal equilibrium model MEMI, which is a comprehensive evaluation index. At present, European cities and Hong Kong mainly use PET indicators to measure outdoor thermal comfort. The latter introduces the three-dimensional microclimate model ENVI-met, which can be used to simulate the unique microclimate system model generated by various unused surfaces and shelters in urban areas, focusing on microscale numerical simulation of the internal surface-plant-air interaction of urban structures, especially feedback between artificial surfaces such as buildings and vegetation in street canyons, backyards, or green spaces [27]. Ariane Middel et al. used ENVI-met to investigate the effects of urban morphology and landscape types on the mid-afternoon microclimate of arid Phoenix, Arizona [16]. Mohammad Taleghani et al. simulated the outdoor microclimate with ENVI-met and converted the data into physiological equivalent temperature (PET) through RayMan to study the influence of urban morphology on urban microclimate and thermal comfort environment [28].

## 4. Hot Spot Analysis of Urban Microclimate Research

### 4.1. Research Hotspots

The bibliographic keywords are the author’s refinement and induction of the main content of the article, and the word frequency analysis of keywords is often used in bibliometrics to reveal the distribution of research hotspots [29]. In the case of a large amount of literature data, the keyword co-occurrence atlas drawn by VOSviewer has the advantages of low label overlap, clear clustering, and good readability [30]. Therefore, VOSviewer is used for keyword co-occurrence analysis in this paper to present the distribution characteristics of foreign urban microclimate research hotspots. In VOSviewer, the minimum threshold of word frequency statistics is set as 10, and the top 100 high-frequency keywords are selected to draw the label view of co-occurrence of keywords in urban microclimate research (Figure 5).

In Figure 5, each circle represents a keyword node, while the size of the circle represents the level of word frequency. The larger the circle, the higher the word frequency, and the more representative the main research content. The connection between the nodes represents the co-occurrence relationship between them, according to which different research clusters are formed. Nodes with the same color belong to the same cluster. Overall, the term “outdoor thermal comfort” had the largest node and the highest frequency, followed by “management”. Outdoor thermal comfort and management constitute the two core concepts of urban microclimate research. Centering on the research topic of urban microclimate, other high-frequency keywords based on co-occurrence relationships present the following four main research clusters, which can be used to analyze the main research contents, perspectives, and methods.

(1) High-frequency words within the green cluster include outdoor thermal comfort, perception, orientation, human thermal comfort, thermal sensation, etc., showing the attention to human comfort in the field of urban microclimate. Research methods for human thermal comfort usually use questionnaires, and the characterization of thermal stress in outdoor spaces usually uses indicators such as Physiologically Equivalent Temperature (PET) and predicted mean vote (PMV). Urban microclimate is an important factor affecting urban thermal environment, and the comfort degree of urban thermal environment directly affects the quality of life and physical and mental health of urban residents. One representative project is RUROS, a project for the Urban Realm and Open Space Revival in Europe led by Marialena Nikolopou and others. The study, which focused on the environment and comfort conditions of urban open spaces, obtained nearly 10,000 questionnaires and human-monitored results from 14 different case study sites in five different countries in Europe, confirming the significant impact of urban microclimate on comfort conditions [31]. Later, Marialena Nikolopou published an article on the relationship between the urban microclimate and outdoor space function in 2007, further discussing the importance of thermal comfort environment under the influence of urban microclimate for urban residents’ life, work, and behavior [32].

(2) The high-frequency words within the red cluster include management, tree species, etc. It shows the correlation between urban landscape, urban forest, sustainable development, and the urban microclimate. The research involves tree species, conservation, habitat, and so on. These keywords indicate that scholars’ research on improving urban microclimate strategies using trees is crucial in dense urban environments. This is not only because of their aesthetic value but also due to their cooling effect during hot periods, which directly affects the local microclimate [33,34]. Carola Helletsgruber et al. revealed in their study that tree species have a great influence on the cooling effect of urban residents and emphasized that tree species in urban areas must be appropriate to the urban climate and site conditions [35]. Urban green space management can offset the negative impact of vegetation area reduction [36]. Research by Carmela Apreda et al. shows that not all residences need more vegetation cover to achieve significant temperature reduction. The the design of green solutions must consider background characteristics: combining knowledge of microclimate processes with site-specific design is critical to optimizing urban implementation and management [37]. Therefore, estimates by policymakers and tree management practitioners are very favorable when planning and managing urban green spaces to improve the availability and delivery of ecosystem services, contributing to urban tree planning to improve urban microclimates and optimize management costs [38].

(3) The high-frequency words within the blue cluster include street canyon, energy Demand, aspect ratio, air flow (V), energy use, etc., which mainly study the energy use in specific city microclimates. For example, Ruwaa Bahgat et al. studied how to reduce the solar radiation received by buildings and street canyons so as to reduce the cooling source used in buildings [39]. Most research methods are energy simulation, such as ENVI-MET, SOLWEIG, TownScope, RayMan, etc.

About 75% of global energy use occurs in urban areas. The street can be seen as an interface between the architectural and urban scales, as it consists of a shared surface between the building and the open urban canopy. Therefore, streets affect both indoor and outdoor microclimates, thus affecting people’s thermal sensation and the energy consumption of urban buildings [40]. Urban streets, which usually cover more than 1/4 of the urban area [41], are important places for people’s outdoor activities in the city [42]. Studies on the combination of streets and microclimates began in the 1970s [43]. Due to people’s continuous attention to the concept of living environment and sustainable development, the research on urban microclimate is more and more extensive, and the research on urban street space and ecological environment is attaining more depth [44]. As the basic unit of urban outdoor space, the environmental quality of urban streets directly affects the comfort of pedestrians in the streets and the physical environment of urban areas and plays a very important role in regulating regional microclimate [45]. Oke et al. suggested that street design could find a “compatibility zone” that would balance the street’s protection from the sun in summer with its demand for solar energy in winter [46]. Therefore, further investigation is needed to provide quantitative information on the best street forms to regulate the urban microclimate and improve climate comfort [40]. At present, the research direction of street canyons mainly focuses on street aspect ratio, airflow, temperature, traffic pollutants, and so on.

(4) The high-frequency words within the yellow cluster include land surface temperature (LST), URBAN heat island intensity (W Intensity), etc., indicating the correlation between urban microclimates and urban heat islands. Land use is analyzed mainly through image data such as TM and TIRS, to study urban microclimates and the urban heat island effect. Temperature is one of the microclimate parameters that distinguishes urban areas, and it is related to the urban heat island effect [47]. A health risk assessment study in the UK found that every 1 °C increase in temperature would increase the mortality rate by 2.1% [48]. The urban heat island is an urban microclimate with extremely high temperature, especially in urban centers, industrial areas, and densely populated areas [49,50,51]. There are three main types of measurement methods for urban heat island strength: field measurements, remote sensing observation methods, and modeling methods [51]. For example, Lin Zhongli and Xu Hanqiu used a combination of Landsat series satellite remote sensing image data and GIS information to draw urban heat island maps of 4 old and new “furnace cities” in China [52].

### 4.2. Frontiers of Research

The term research frontier is an information science concept, first proposed by bibliometrist Price in 1965 and used to describe the dynamic nature of the research field [53]. The identification and tracking of research frontiers can provide researchers with the latest evolutionary dynamics of discipline research, predict the direction of discipline development and research hotspots, and is of great significance to the anti-war construction of disciplines. CiteSpace can use burst detection algorithms to identify research frontiers. The basic method is to detect the words with a high-frequency change rate from a large number of subject words through the time distribution of keywords and rely on the trend of word frequency change, not just the level of frequency, to determine the frontier field and development trend [54]. Through the analysis of the function of CiteSpace’s sudden test, it is concluded that the research frontiers in the field of urban microclimate include building energy consumption, ecosystem services, and parks.

#### 4.2.1. Urban Microclimate and Building Energy Consumption

Urban microclimates are characterized by changes in outdoor climate, surface temperature, humidity, wind speed, and wind direction [55]. Some human factors can lead to urban microclimates. Existing studies have shown that the local microclimate of a building has an important influence on building energy consumption. In contrast, the urban microclimate is comprehensively influenced by a variety of factors such as geometric characteristics, layout mode, and the physical properties of the underlying surface of the street (residential) area. Thus, it has an important influence directly or indirectly on building energy consumption and the indoor thermal environment [56]. The urban heat island phenomenon in temperature further exacerbates urban environmental and human health challenges and adds to the burden of existing urban systems, with urban energy consumption soaring due to increased cooling needs [57,58,59]. Chiru Chang et al. estimate that five major U.S. cities would increase peak electricity demand by 2–4 percent once temperatures exceeded 15–20 °C. In the UK, building energy accounts for more than half of total energy consumption (41% in the EU and 36% in the US) [60]. Zhenhong Tian et al.’s study shows that urban microclimate has a significant impact on building energy use [61]. The study used building models to simulate 2017 San Francisco weather data to assess the impact of urban microclimate on building energy use. The results show that the impact of urban microclimate on energy use in air conditioning systems is significant, and the use of different microclimate data can lead to an annual difference of more than 100% in heating energy use and an annual difference of up to 65% in cooling energy use. Gunwon Lee and Yunnam Jeong studied the relationship between urban microclimates and the variables of the surrounding natural environment and building energy consumption. They found that the larger the wind speed, the smaller the energy consumption, and the higher the temperature and humidity, the greater the energy consumption [62]. It can be seen that urban microclimates can strongly influence the use, demand, and thermal resilience of buildings for energy [61]. Therefore, in recent years, how to quantitatively predict and evaluate the impact of urban microclimate on building energy consumption, establish the ability to resist urban microclimate change, and provide a scientific basis for the planning and architectural design of street (residential) district is an urgent subject to be strengthened.

#### 4.2.2. Urban Microclimate and Ecosystem Services

Over the past few decades, rapid urbanization has produced built environments with high energy consumption and severe vegetation destruction, major providers of urban ecosystem services and play an important role in regulating the local microclimate [63]. Urban microclimates are heavily influenced by urban design, with complex interactions between outdoor conditions (e.g., temperature, speed, wind speed, solar radiation) and the morphological parameters that shape cities (e.g., urban density, building forms, roads, streets, and canopy geometries, building orientation). Further, ecosystem services can be used to develop urban design parameters and standards for planning and projects to mitigate heat waves and high temperatures caused by urban heat island effects [64]. Urban ecosystem services can be provided patches of vegetation, such as urban forests, wetlands, and parks [65,66,67]. Nicholas H. Wolff et al. found through an extensive social survey of nearly 500 villages that urban forests play an important role in regulating local microclimate [68]. Planners and decision-makers are becoming aware of the importance of urban ecosystem service. They seek the most appropriate configuration of urban green vegetation to improve the urban microclimate, such as three-dimensional green structures, green roofs, and street tree species [65,69,70]. People are now focusing on the use of natural ecosystems to find solutions to various urban problems and improve the urban microclimate [63]. The New York Times had an article on 10 June 2021 titled “Our Response to Climate Is Missing Something Big, Scientists Say” [71]. The article states, “the world needs to treat warming and biodiversity loss as two parts of the same problem”. Such explicit recognition that climate and ecology study the same system would make effective policy decisions more likely [71]. At the same time, microclimate regulation, as an important urban ecosystem service, is often used by researchers as an indicator to test the value of ecosystem services [72,73,74].

#### 4.2.3. Urban Microclimate with Urban Parks

The city park is the main place for most residents to rest, engage in activities and communicate and is one of the main symbols of urban vitality. Therefore, cities should pay attention to creating microclimates in parks to provide residents with healthy and comfortable rest and social space. In addition to the basic microclimate parameters such as temperature, wind speed, relative humidity, and solar radiation intensity in urban parks, different spatial enclosures and landscape elements also affect the microclimate environment [6]. Microclimate comfort is an important factor for people to choose a park and, to a certain extent, determines the quality of citizens’ recreation. Diana E. Bowler et al. used a meta-analysis to synthesize data on the cooling effects of parks, and the results showed that, on average, park temperature decreased by 0.94 °C during the day and that larger parks and parks with trees may be cooler during the day [22]. Most studies on the thermal performance of urban parks have taken a simulation approach and investigated how different configurations of landscape elements (i.e., vegetation, pavement, etc.) and vegetation types (deciduous and evergreen trees, shrubs, and turf) affect the thermal performance of parks [75,76,77,78,79]. However, it is unclear whether there are stable temperature or humidity gradients in urban parks. The dominant factors for the microclimate in urban parks have not been identified and further research is needed. However, with the reduction of urban green space, the cemetery, as a public urban green space, can also provide a healthy and comfortable space for residents to rest and communicate. Analytical studies can be conducted linking urban parks to cemeteries.

## 5. Conclusions and Outlook

Since the beginning of the 21st century, urban microclimate research literature continues to increase, and the research results have attracted the attention of urban planning, management, and ecological environment departments, which have important reference significance for urban management and planning. In this paper, based on the core database of Web of Science (WoS) and through using the literature metrological analysis software CiteSpace and VOSviewer, literature metrological analysis software, literature related to urban microclimate is studied to understand the historical track of the urban microclimate field and help predict the future research direction.

The results show that over the past 30 years, research on urban microclimates has become more diverse and global. Worldwide, scholars in China, the United States, Italy, and Germany are the most important contributors to the literature on urban microclimates. Today, the publication rate of Chinese researchers is the highest in the world, and the Chinese Academy of Sciences is the second most productive research institution. In addition, the collaborative network of institutions and authors shows that the influence of European and American countries in the field of urban microclimate has always been central, while the academic influence of China is still insufficient. Keyword analysis was an effective method to identify hot issues and research frontiers. The analysis shows that research on urban microclimates mainly focuses on thermal comfort, strategies to improve urban microclimates, urban street canyons, and the urban heat island effect. The most advanced research addresses three points: correlational studies of urban microclimate and building energy; mechanism studies on how ecosystem services improve urban microclimate; research on the interaction between urban parks and urban microclimate.

## Figures and Tables

**Figure 1 ijerph-19-04741-f001:**
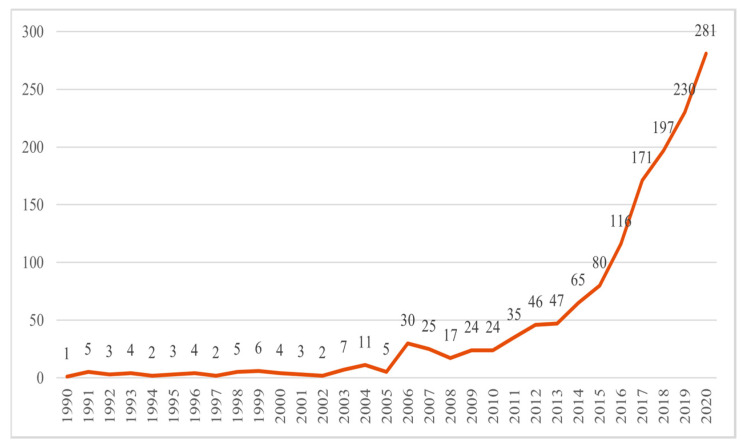
Time distribution of the number of artical posts.

**Figure 2 ijerph-19-04741-f002:**
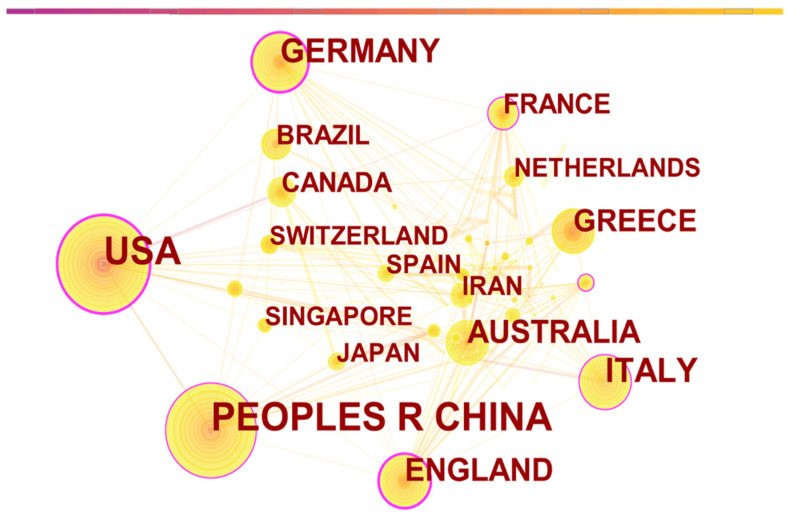
Country Cooperation Network Map.

**Figure 3 ijerph-19-04741-f003:**
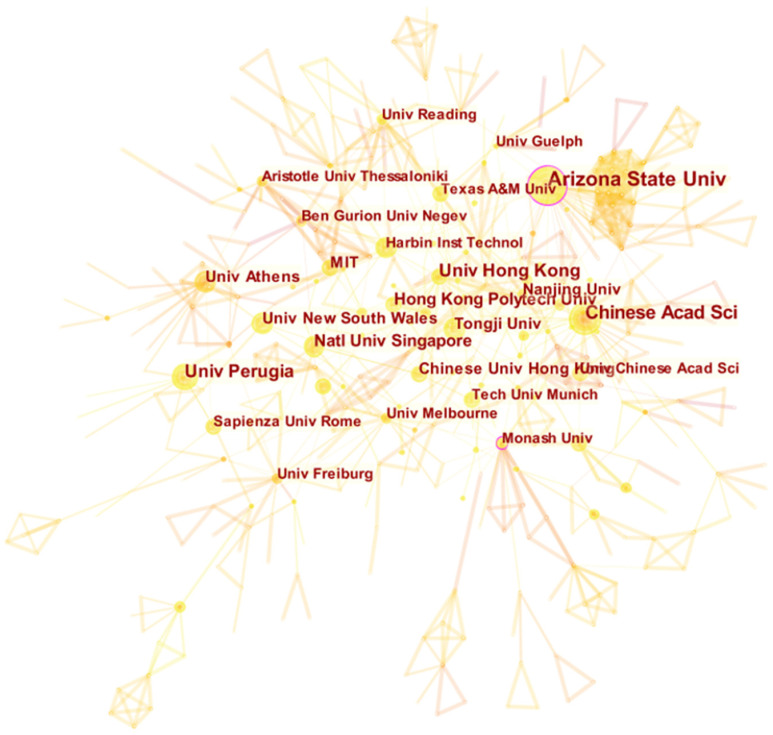
Institutional cooperation network map.

**Figure 4 ijerph-19-04741-f004:**
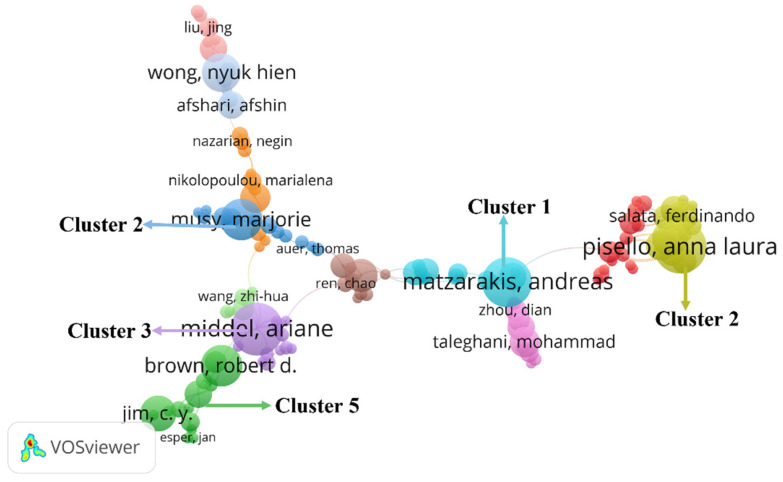
Author Co-cited Knowledge Graph.

**Figure 5 ijerph-19-04741-f005:**
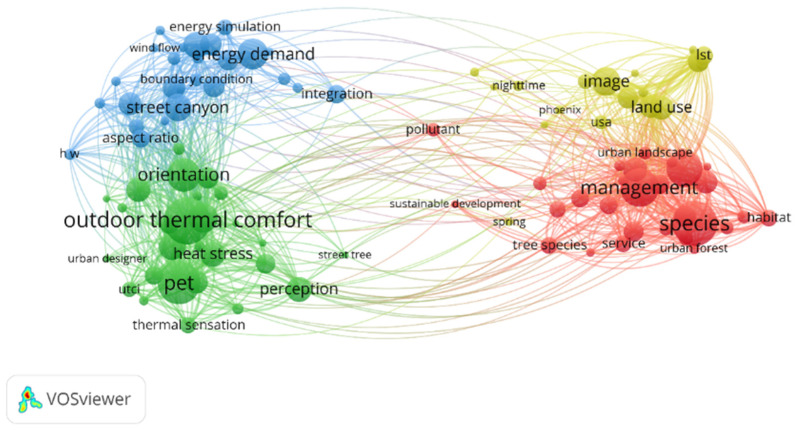
Keyword co-occurrence tag view of domestic and foreign urban microclimate research.

**Table 1 ijerph-19-04741-t001:** Top 10 countries by publication post.

Country	Publications	Centrality	Country	Publications	Centrality	Publications
PEOPLES R CHINA	311	0.14	AUSTRALIA	102	0.04	311
USA	279	0.32	GREECE	85	0.02	279
ITALY	127	0.11	FRANCE	56	0.18	127
GERMANY	125	0.30	CANADA	53	0.08	125
ENGLAND	102	0.24	BRAZIL	48	0.05	102

**Table 2 ijerph-19-04741-t002:** Top 10 Institutions by Post.

Institution	Publications	Centrality	Institution	Publications	Centrality
Arizona State University	47	0.18	Hong Kong Polytech University	23	0
Chinese Academy of Sciences	34	0.08	Tongji University	21	0.04
University Hong Kong	25	0.1	University New South Wales	20	0.04
University Perugia	25	0.01	University Athens	19	0.06
Natl University Singapore	24	0.04	Chinese University Hong Kong	18	0.08

**Table 3 ijerph-19-04741-t003:** Key authors with more than nine publications.

Name	Number
Ariane Middle	16
Anna Laura Pisello	15
Jan Carmeliet	12
C. Y. Jim	12
Jonas Allegini	10

**Table 4 ijerph-19-04741-t004:** Top 20 cited publications based on published literature on urban microclimate.

No.	Year	Author	Journal	Country/Insititute	Topic	Citations
1	2010	Bowler, D.E. et al.	LANDSCAPE AND URBAN PLANNING	Wales/Bangor University	Greenery	1038
2	1999	Matzarakis, A. et al.	INTERNATIONAL JOURNAL OF BIOMETEOROLOGY	Germany/University Freiburg	PET	535
3	1998	Bruse, M.; Fleer, H.	ENVIRONMENTAL MODELLING & SOFTWARE	Germany/Ruhr University Bochum	ENVI-met	487
4	2006	Ali-Toudert, F.; Mayer, H.	BUILDING AND ENVIRONMENT	Germany/University Freiburg	Outdoor thermal comfort	478
5	2008	Alexandria, E.; Jones, P.	BUILDING AND ENVIRONMENT	Wales/Cardiff University	Outdoor thermal comfort	451
6	2012	Ng, E. et al.	BUILDING AND ENVIRONMENT	Peoples R China/Chinese University Hong Kong	Urban design; Steeet canyon	401
7	2006	Fisher, J.I. et al.	REMOTE SENSING OF ENVIRONMENT	USA/Brown University	Landsat; Greenery	350
8	2006	Nikolopoulou, M.; Lykoudis, S.	BUILDING AND ENVIRONMENT	England/ University Bath; Greece/Natl Observ Athens	Outdoor thermal comfort	334
9	2003	Dimoudi, A.; Nikolopoulou, M.	ENERGY AND BUILDINGS	Greece/Ctr Renewable Energy Source	Greenery	330
10	2000	Carlson, T.N.; Arthur, S.T.	GLOBAL AND PLANETARY CHANGE	USA/Penn State University	Microclimatic variables	325
11	2007	Ali-Toudert, F.; Mayer, H.	SOLAR ENERGY	Germany/University Freiburg	Street canyon	301
12	2007	Chang, C.R. et al.	LANDSCAPE AND URBAN PLANNING	Taiwan/Chinese Culture University	Urban park	290
13	2013	Lovell, S.T.; Taylor, J.R.	LANDSCAPE ECOLOGY	USA/University Illinois	Management	275
14	2014	Middel, A. et al.	LANDSCAPE AND URBAN PLANNING	USA/Arizona State University; Germany/ University Kaiserslautern	Urban form and design	268
15	2006	Johansson, E.	BUILDING AND ENVIRONMENT	Sweden/Lund University	outdoor thermal comfort	261
16	2003	Steemers, K.	ENERGY AND BUILDINGS	England/University Cambridge	Energy demand	259
17	2003	Dousset, B.; Gourmelon, F.	ISPRS JOURNAL OF PHOTOGRAMMETRY AND REMOTE SENSING	USA/University Hawaii; France/UBO	Urban landscape; Summertime microclimate	251
18	2007	Jenerette, G.D. et al.	LANDSCAPE ECOLOGY	USA/Ohio State University	Surface temperature; Greenery	247
19	2015	Taleghani, M. et al.	BUILDING AND ENVIRONMENT	Netherlands/Delft University Technology	Outdoor thermal comfort	242
20	2011	Matzarakis, A. et al.	INTERNATIONAL JOURNAL OF BIOMETEOROLOGY	Israel/Ben Gurion University Negev	Greenery; Outdoor thermal comfort	240

## Data Availability

The Web of Science (WoS) data can be accessed through the WoS’s official website: https://www.webofscience.com/wos/alldb/basic-search (accessed on 23 September 2021).

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
