# Peer review of "A Review of Urban Microclimate Research Based on CiteSpace and VOSviewer Analysis"

_ijerph, 2022, doi:10.3390/ijerph19084741_

Round 1

Reviewer 1 Report

The authors have provided an interesting systematic review of the urban microclimate literature and I commend them for their analysis and discussion of the topic.  Nice job! This review article may save other researchers considerable time and thus meets the stated purpose of the paper.

  1. There are numerous editorial suggestions in the attached document.
  2. Line 55: Climate does not adapt to people. It may change in response to human activity and pollution but it does not adapt.  People are capable of adaptation and may change behaviors in response to climate change.  This may be a simple word choice issue.
  3. Introduction and Methods: There are some methods included in the introduction (lines 61` through 64). Incorporate this statement in Methods. There is a purpose statement in Methods (Lines 95-97) that should be incorporated in the last paragraph of the Introduction. 
  4. The authors identified 1893 publication and analyzed 1455 articles. What happened to the other 38?  Tell the reader why these were excluded.
  5. Figure 1: The number of articles is placed on the curve and are not particularly legible.  Place each number below the year on the x axis or adjacent to the curve.
  6. There are a number of terms used that are not clear to the reader. Define “cooperation network” before it is used in line 122. It is unclear whether any of these entities are cooperating with each other from this analysis.
  7. Line 122 refers to a “national network”. What nation??? This paper evaluates papers from across the world. Define what you mean by national network or select another term.
  8. Figure 2. There is some illegible text in the upper right hand corner of Figure 2.  What is it.  Either make it legible or delete.
  9. Figure 2 is labeled as a “National Cooperation Network.” How are these nations cooperating? How is this evident in this figure? What topics are they cooperating on?
  10. Table 1: What is “centrality”.  Define prior to use.
  11. There are two Table 2s. One table 2 is on page 4-5 and the second is on page 6-7. Resolve.
  12. Figure 3 is too small and font is illegible. It also has the same illegible text in the upper left hand corner.  Either make this text readable or delete. 
  13. Figure 4. Place circles around each cluster and label. Explain how you delineate the boundaries of each cluster.
  14. Second Table 2 (or Table 3???)Add a column to the table. Describe in 1 or two words the topic covered in each of the 20 papers listed. You may come up with new designations or use the terms associated with the colored clusters of Figure 5 (lines 248-322)
  15. Figure 5 is not legible and needs to be larger. The key word division is really interesting.
  16. Lines 356-362: Is this concept generalizable to all climate regimes or is this specific to one geographic region?
  17. Section 4.2.3: Note there has been considerable discussion of cemeteries in urban settings acting as “park land” or green space. See https://doi.org/10.1016/j.ufug.2021.127078.  You might want to include this green space in your discussion.
  18. There are no acknowledgements therefore delete this item (lines 443-445)
  19. Align references.

Reviewer 2 Report

The authors presented the bibliometric analysis of recent urban microclimate research using the Web of Science database and supported by the analysis with two software.

The considerations are as follows:

-There are a few typos, spelling, or reference mistakes as the name Ooka in line 52. Therefore, it is suggested that the authors review the manuscript carefully.

-The equation in line 164 should be separated from the main text.

-The data from lines 168 and 169 could also be presented as a table.

-Is the reviewer's opinion that the title should indicate the purpose of the manuscript instead of displaying the name of the software. The software was used just two times each. So, placing the name of the software in the title seems to be too much. It would be better to indicate where the recent studies are aiming. 

Reviewer 3 Report

This paper deals with the topic of urban microclimate and quality of life of urban residents.  It is well worth pursuing further but I found that this paper only represents a start to such a program and I think that more work needs to be done to completely clarify the issues. If the authors can better explain the context of the larger project that encompasses this idea in the introduction to the work, that would be great.
